# Quantifying intervals to diagnosis in myeloma: a systematic review and meta-analysis

Constantinos Koshiaris,[1] Jason Oke,[1] Lucy Abel,[1] Brian D Nicholson,[1] Karthik Ramasamy,[2,3] Ann Van den Bruel[1]

This study has been presented as an oral presentation at the SAPC Conference, Dublin, July 2016.

[1]Nuffield Department of Primary Care Health Sciences, University of Oxford, Oxford, UK
[2]Department of Haematology, Oxford University Hospitals NHS Trust, Oxford, UK
[3]National Institute for Health Research (NIHR) Biomedical Research Center Blood Theme, Oxford, UK

**Correspondence to**
Mr Constantinos Koshiaris;
constantinos.koshiaris@phc.ox.ac.uk

## ABSTRACT

**Objectives** To quantify the duration of each step of the diagnostic pathway for patients with multiple myeloma from symptom onset to confirmation of diagnosis.

**Design** Systematic review and meta-analysis.

**Data sources and selection criteria** The MEDLINE and Embase databases were searched up until January 2018 to identify articles that reported time intervals from onset of symptoms to diagnosis. Articles focusing on children or adolescents and on the asymptomatic form of the disease (monoclonal gammopathies and smouldering myeloma) were excluded.

**Data collection and data analysis** Data were extracted independently by two reviewers. Weighted estimates of the median and IQR were calculated. Risk of bias was assessed using the Aarhus checklist.

**Main results** Nine studies were included. The patient interval (first symptom to first presentation) had a median of 26.3 days (IQR: 1–98, n=465, two studies). Subsequently, the primary care interval (first presentation to first referral) was 21.6 days (IQR: 4.6–55.8, n=326, two studies), the diagnostic interval (first presentation to diagnosis) was 108.6 days (IQR: 33.3–241.7, n=5395, seven studies) and the time to diagnosis (first symptom to diagnosis) interval was 163 days (IQR: 84–306, n=341, one study). No studies reported data for the referral to diagnosis interval.

**Conclusion** The review demonstrates that there is scope for significant reductions in the time to myeloma diagnosis. At present, many patients experience a diagnostic interval longer than 3 months until diagnosis is confirmed.

**Review registration** Not available. Protocol available in the appendix.

## INTRODUCTION

Myeloma is a haematological malignancy characterised by uncontrolled plasma cell production in the bone marrow. It was the 17th most common cancer in the UK in 2013 accounting for 2% of all new cancer cases. Currently, there are more than 17 500 patients with myeloma in the UK with approximately 5500 cases being diagnosed every year.[1 2] It is a cancer that mainly affects the elderly population with 59% of the patients being diagnosed over the age of 70 years[2] and with a 5-year survival of 47%.[3]

### Strengths and limitations of this study

► First systematic review to quantify the whole diagnostic pathway for patients with multiple myeloma including the different intervals in each step of the pathway.

► Use of all available information including the IQR rather than focusing on measures of central tendency like the mean and the median.

► No universally accepted methods for formal meta-analysis of median and IQR.

► Limited number of studies reporting the patient and primary care intervals, and no studies reported the referral to diagnosis interval, so any inferences regarding the referral to diagnosis interval should be interpreted with caution.

It is considered one of the hardest cancers to suspect in primary care. Symptoms of myeloma are very common and non-specific, such as back pain, bone pain, fatigue and repeated infections.[4] This in combination with the fact that myeloma is a very rare condition in primary care results in very low predictive values for individual symptoms. For example, primary care patients with back pain, one of the most common myeloma symptoms, only have a 0.1% risk of myeloma.[5] By comparison, patients with rectal bleeding have a 2.4% risk of colorectal cancer.[6]

As a result, half of patients with symptomatic myeloma have three or more consultations in primary care before they are referred to specialist care, which is more than in any other cancer.[7] Attributing symptoms to comorbidities further prolongs the diagnostic process, which is particularly relevant in this older age group.[8 9]

Delays in diagnosing myeloma allow complications to develop (end organ damage), such as pathological fractures, irreversible renal failure and in some cases spinal cord compression.[10–12] These are considered medical emergencies in their own right and limit the opportunity for initiating effective

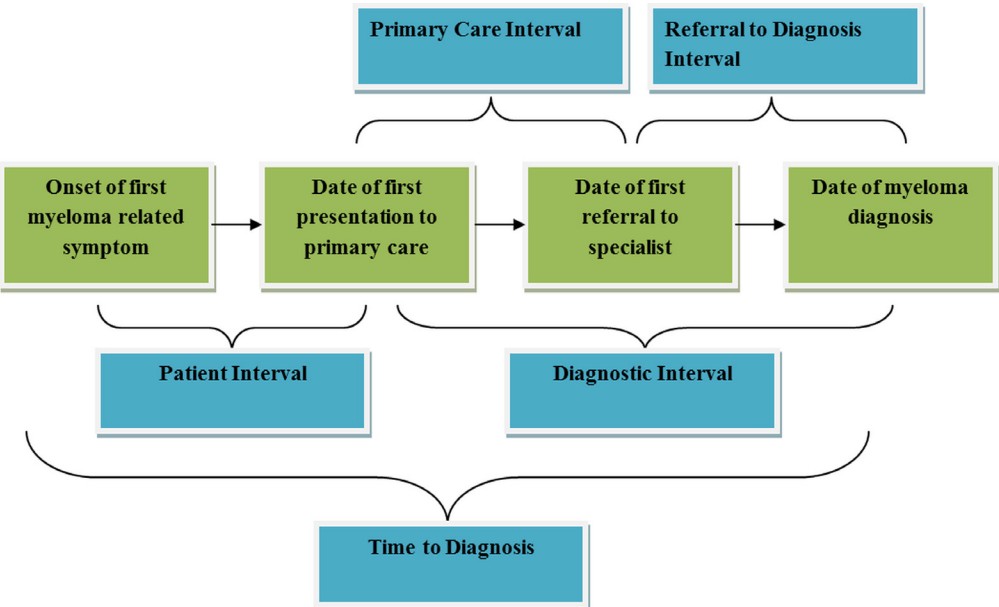

**Figure 1** Outcome definition.

treatment.[13] A delayed diagnosis is also linked with higher cancer stage,[14 15] which is in turn associated with poorer survival.[16] Patients with longer diagnostic intervals also experience shorter disease free survival and more complications from treatment.[14]

Quantifying the time intervals leading up to diagnosis is important as it will inform future interventions that aim to shorten the diagnostic process. The aim of this systematic review was to quantify each step of the diagnostic pathway to myeloma diagnosis and identify where to focus efforts to reduce diagnostic delay.

## METHODS

A protocol is available in the online supplementary appendix A1. A copy of the search strategy can be seen in the online supplementary appendix A2. We searched Embase and MEDLINE until January 2018 for studies that quantified any or all of the following five intervals[17]: the patient interval (from symptom onset to first consultation); the primary care interval (from first consultation for that symptom to referral to secondary care); the diagnostic interval (from first consultation with a myeloma-related symptom to diagnosis) and the time to diagnosis (from symptom onset to diagnosis). In addition, we looked for studies that estimated the referral to diagnosis interval (figure 1). Citation searching of key references like the Aarhus statement was conducted, and we also searched the reference list of systematic reviews with similar research questions.[18 19] We included any study designs that quantified at least one of the intervals mentioned above in days or months. Studies reporting the length of an interval only in number of consultations or referrals were excluded as were studies focusing on children or adolescents (<18 years) and on the asymptomatic forms of the disease (monoclonal gammopathies

and smouldering myeloma). We included papers with an abstract in English but did not exclude full-text articles based on language (as long as there was an English abstract). Conference abstracts were excluded. Two reviewers (CK/LA) selected papers for inclusion using the criteria listed above, on title and abstracts first and on full text second. Disagreements were resolved through discussion with a third reviewer (JO/AVdB).

### Data extraction

Two reviewers (CK/LA) independently extracted data from the included studies into a predefined spreadsheet. Study characteristics including author, year of publication, country of data collection, type of study, myeloma-related symptoms and sample size were extracted, as well as descriptive statistics including median, IQR, range, mean and the SD for each interval. Authors were contacted if data were not available or not in the appropriate format for extraction (ie, categorical rather than continuous).

### Risk of bias assessment

The risk of bias was assessed by two independent researchers (CK/LA) using the Aarhus checklist.[18] The Aarhus checklist is a 20-item tool designed to help researchers design and evaluate studies on early diagnosis of cancer. It examines studies in terms of acknowledgement of the different biases influencing time point measurement and interval definition.

### Analysis

In the context of illness duration, intervals are usually not normally distributed; therefore, we used the median and IQR to summarise the data. We present the 25th, 50th and 75th percentiles for all intervals. For intervals reported by more than one study, the pooled estimate was calculated by taking a weighted mean for each percentile. The

**Table 1** Study characteristics

| Study design | Study period | Population characteristics (age and gender) | Patients with myeloma (total) | Outcome measure (interval) |
|---|---|---|---|---|
| Friese et al (USA)[8] | | | | |
| Retrospective analysis | 1992–2002 | Mean age 76.3 years 46% males | 3831 | Diagnostic |
| Howell et al (UK)[31] | | | | |
| Survey | 2004–2011 | Median age 69.9 years 66.9% males | 341 | Patient diagnostic Time to diagnosis |
| Lyratzopoulos et al (UK)[26] | | | | |
| Audit data | 2009–2010 | Not reported | 176 | Primary care |
| Varga et al (Hungary)[29] | | | | |
| Retrospective analysis | Not reported | Median age 60 years 50% males | 193 | Diagnostic |
| Neal et al (UK)[25] | | | | |
| Retrospective analysis | 2001–2002 | Mean age 72 years 53% males | 221 | Diagnostic |
| Din et al (UK)[24] | | | | |
| Retrospective analysis | 2007–2010 | Median age 72 years 56% males | 500 | Diagnostic |
| Lyratzopoulos et al (UK)[27] | | | | |
| Audit data | 2009–2010 | Not reported | 124 | Patient |
| Goldschmidt et al (Israel)[30] | | | | |
| Retrospective analysis | 2002–2011 | Median age 63 years 53% males | 107* | Diagnostic |
| Swann et al (UK)[28] | | | | |
| Audit data | 2014 | Not reported | 202 | Primary diagnostic |

*The total sample size for this study was 110 patients out of which seven were diagnosed with plasmacytoma. The analysis was conducted on 107 patients that had complete data.

weight was obtained by dividing the sample size in each study with the total numbers of patients. We also fitted a distribution through the three weighted percentile estimates where appropriate in order to generate the shape of the distribution of the interval under investigation. We chose the lognormal distribution as time intervals are usually skewed to the right.[4] A prespecified sensitivity analysis was conducted by excluding the study with the higher risk of bias. The sensitivity analysis was conducted only for the diagnostic interval as the rest of the outcomes were reported by only one or two papers, which can be seen in table 1.

### Patient and public involvement
No patients or public were involved in this study.

## RESULTS
We identified 3343 citations from the Embase and MEDLINE searches. After removal of conference abstracts and duplicates, we screened 1271 titles and abstracts, and 16 studies were candidate for inclusion. Nine studies were included in the final analysis (figure 2). Seven papers in total were excluded for the following reasons: reported the duration in numbers of consultations (n=2)[5 7]; none of the prespecified outcomes were reported (n=1)[20]; reported the same outcome based on the same database as one of the other included papers, so inclusion of this paper in the data synthesis would result in double counting the same patients (n=1)[21]; data were not in an appropriate format (n=2)[14 22]; and because the interval under investigation was reported only for patients that were referred to very specific departments making it a very selective population compared with the other studies (n=1).[23]

### Study characteristics
A summary of all the included papers is provided in table 1. Studies were published between 2009 and 2018, and the sample size ranged from 107 to 3831 patients. Five studies reported intervals in various cancers,[24–28]

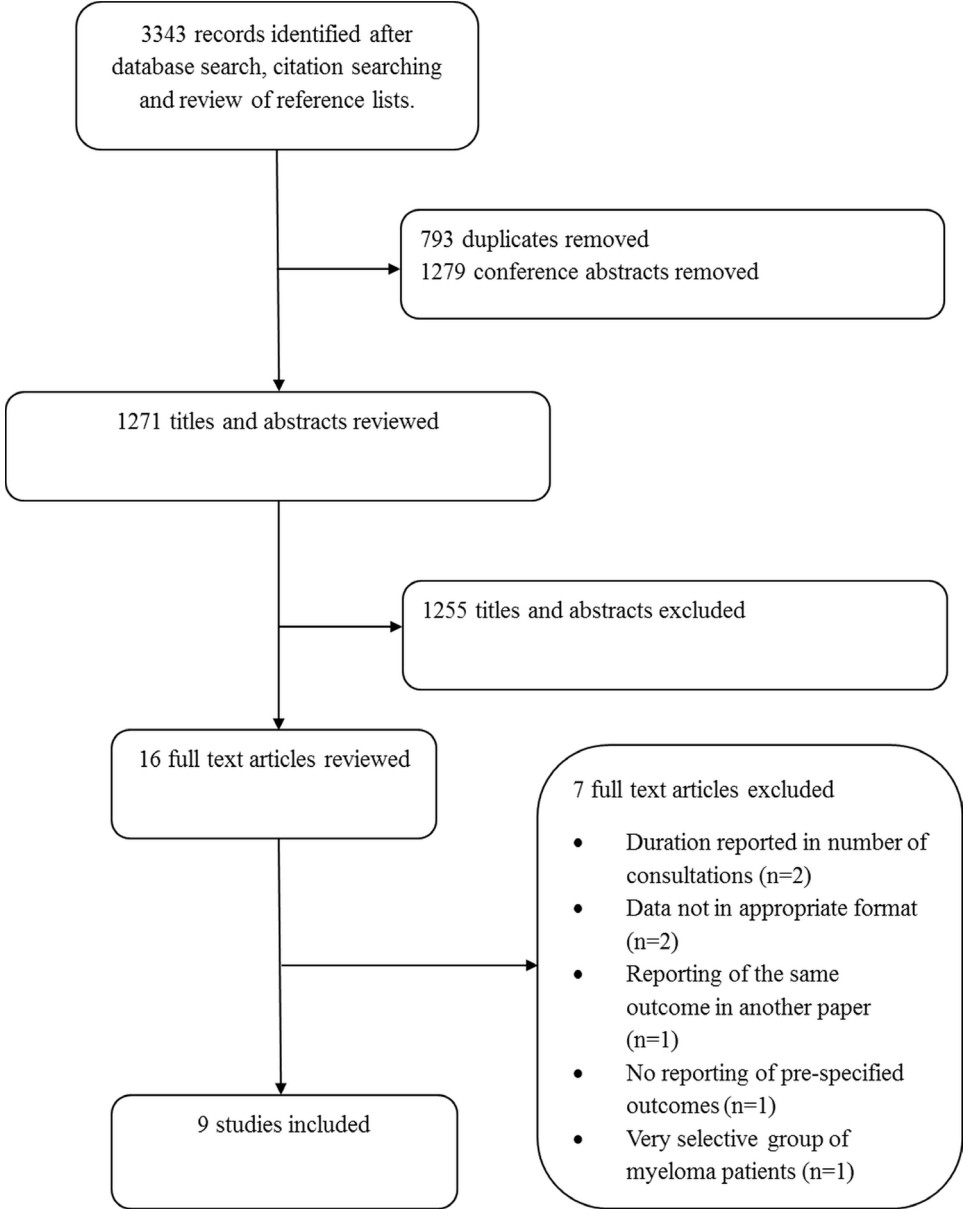

**Figure 2** Study selection flow chart.

three reported only myeloma[8 29 30] and one for haematological malignancies.[31]

Six studies were conducted in the UK,[24–28 31] one in the USA,[8] one in Hungary[29] and one in Israel.[30] Two UK studies used data from two separate Clinical Practice Research Datalink (CPRD) cohorts,[24 25] a database of routinely collected electronic primary care records. Two other UK studies used data from the English National Audit of Cancer Diagnosis in Primary Care 2009–2010[32]; we extracted primary care interval data from the larger study and the patient interval from the smaller study.[26 27] Another UK study used data from the English National Audit of Cancer Diagnosis in Primary Care 2014, and the last UK study was a patient survey on patients diagnosed with haematological malignancies.[28 31] The study conducted in Hungary analysed data collected from patients treated in a haematology centre, and the study

conducted in the USA analysed a retrospective database collected from the Surveillance, Epidemiology, and End Results programme (SEER).[8 29] The study conducted in Israel used data from the Israeli health maintenance organisation linked with the Israel National Cancer Registry.[30]

### Definition of intervals to diagnosis

There was substantial heterogeneity in the symptoms and time points used to define each interval (table 2). In total, 19 different symptoms were used to define the start of myeloma but studies varied greatly regarding which symptom (or symptoms) were used, ranging from 3 to 12 symptoms. Three studies did not report the starting symptoms.[26–28] Some studies included multiple symptoms in more general categories.[24 25 31] For example, Howell *et al* used a general pain category that included

**Table 2** Symptoms and date definitions

| Symptoms used | Onset of first symptom | Date of first presentation in healthcare services | Date of first referral | Date of diagnosis |
|---|---|---|---|---|
| **Friese et al[8] (USA)** | | | | |
| Anaemia<br>Packed red blood cell transfusion<br>Back pain | NA | 1 year before diagnosis | NA | SEER cancer diagnosis date |
| **Howell et al[31] (UK)** | | | | |
| Tiredness<br>Pain<br>Shortness of breath<br>Infections<br>Joint problems/fractures<br>Stomach/bowel symptoms<br>Other | Patient reported | Patient reported | NA | Date provided by the haematological malignancy Diagnostic service |
| **Lyratzopoulos et al[26] (UK)** | | | | |
| Not reported | Estimated based on patient's clinical records | 2 years before diagnosis | Date that the referral letter was sent | Clinical records and hospital correspondence |
| **Varga et al[29] (Hungary)** | | | | |
| Bone symptoms<br>Anaemia<br>Renal failure<br>General symptoms<br>Other<br>Tumour presence<br>Metastatic bone disease | NA | 3 years before diagnosis | NA | Tertiary haematology centre |
| **Neal et al[25] (UK)** | | | | |
| Bleeding<br>Bone pain<br>Bruising<br>Anaemia<br>Fatigue<br>Anorexia<br>Weight loss | NA | 1 year before diagnosis | NA | First occurrence of a myeloma Read Code in the patient's record in CPRD database |
| **Din et al[24] (UK)** | | | | |
| Bleeding<br>Bone pain<br>Bruising<br>Anaemia<br>Fatigue<br>Anorexia<br>Weight loss | NA | 1 year before diagnosis | NA | First occurrence of a myeloma Read Code in the patient's record in CPRD database |
| **Lyratzopoulos et al[27] (UK)** | | | | |
| Not reported | NA | 2 years before diagnosis | Date that the referral letter was sent | Clinical records and hospital correspondence |
| **Goldschmidt et al[30] (Israel)*** | | | | |
| Pain (back, cervical spine, musculoskeletal, non-specific)<br>Infection<br>Weight loss<br>Fatigue<br>Peripheral oedema<br>Constipation<br>Presyncope<br>Syncope<br>Dizziness | NA | 2 years before diagnosis | NA | Israel National Cancer Registry |
| **Swann et al[28] (UK)** | | | | |
| Not reported | NA | 2 years before diagnosis | Date that the referral letter was sent | Hospital Episode Statistics |

NA–Not applicable

musculoskeletal, abdominal, chest and other type of pains, while the two CPRD studies included multiple musculoskeletal symptoms under a general bone pain category.[24 25] In addition, studies using CPRD or SEER data were using predefined symptoms to identify the onset of disease, while other studies like Howell *et al* documented the full range of symptoms reported by the patients during this time.

The start of the measuring period was defined as the date of onset of the first symptom or the date of first presentation for a myeloma-related symptom depending on whether the studies were investigating the patient interval, the diagnostic or both. Study authors used various prediagnostic time intervals to identify the first symptom (at 1, 2 or 3 years before diagnosis). Three identified the first symptom at 1 year before diagnosis, three at 2 years and one at 3 years. One study used patient-reported dates. Goldschmidt *et al*[30] did not use the first symptom as the start of the measurement period, but they used the first combination of symptom and laboratory result (ie, the earliest of blood test+pain complaint or two blood tests within a month or two pain complaints within 1–3 months).

### Risk of bias

Most of the studies included in the analysis had a low risk of bias (online supplementary appendix A3 and A4). All studies clearly defined the start and end point of the intervals, and in most cases there was an adequate description of the databases along with the strengths, limitations and biases arising from the definitions of the different intervals and time point. Only one study did not mention the different limitations and biases arising from the study design and the choice of definitions for time points and intervals.[29] Most common sources of bias that were described included recall bias for studies that were using patient reported data and misclassification bias for studies that were using databases like CPRD. Most studies used a theoretical framework to define each interval usually the one reported by Olesen *et al*[17] or the Aarhus statement.[18] The category with the higher risk of bias was the use of a hierarchical rationale to determine the date of diagnosis, that is, date of first histological confirmation of the malignancy or date of admission to the hospital, for example. Most studies mentioned how the date of diagnosis was obtained, but there was no adequate description on how the choice of a particular definition can affect the diagnostic pathway.

### Quantifying intervals

Seven papers reported the diagnostic interval,[8 24 25 28–31] two papers reported the patient interval,[27 31] two papers reported the primary care interval[26 28] and one paper reported the time to diagnosis interval.[31] No studies reported the referral to diagnosis interval. The length of the different intervals can be seen in table 3, and the fitted log normal distributions are shown in figure 3 along with the parameters used to fit them.

For the diagnostic interval, the pooled weighted mean of the 50th percentile is 108.6 days (n=5398) and the IQR is from 33.3 to 241.7 (n=5288). Removing the study with the largest risk of bias[29] based on the Aarhus statement checklist did not alter the results (107.9 days, IQR: 31.3–242.2). An additional sensitivity analysis was conducted by excluding the Goldschmidt *et al*[30] study as it was an outlier, but the results were not affected (pooled median of 103.8). While all the studies reported a median diagnostic interval less than 5 months, this study had an interval of 11.2 months. The IQR was not estimated for this sensitivity analysis as it was not reported by the authors.

The pooled estimate of the 50th percentile of the patient interval is 26.3 days (IQR: 0.7–97.7, n=465). The primary care interval was reported by two studies[26 28] with a median of 21.6 days (IQR: 4.6–55.8, n=326), and the time to diagnosis interval was also reported by one study[31] with a median of 163 days (IQR: 84–306, n=341).

No study reported the referral to diagnosis interval, but it can be inferred by subtracting the median length of the primary care and patient intervals from the diagnostic interval or the time to diagnosis interval. The median length of the referral to diagnosis interval can range from 60.7 to 115.1 days depending on whether we use the diagnostic or the time to diagnosis interval for the inference.

| Table 3 | Length of intervals | | | |
|---|---|---|---|---|
| **Percentile** | **N** | **25th** | **50th** | **75th** |
| **Patient interval** | | | | |
| Howell *et al*[31] | 341 | 1 | 31 | 122 |
| Lyratzopoulos *et al*[27] | 124 | 0 | 13.5 | 31 |
| Weighted estimate | 465 | 0.7 | 26.3 | 97.7 |
| **Primary care interval** | | | | |
| Lyratzopoulos *et al*[26] | 176 | 5 | 21 | 55 |
| Swann *et al*[28] | 150 | 4.2 | 23.5 | 56.8 |
| Weighted estimate | 326 | 4.6 | 21.6 | 55.8 |
| **Referral to diagnosis interval** | | | | |
| No papers reporting this interval | | | | |
| **Diagnostic interval** | | | | |
| Friese *et al*[8] | 3831 | 27 | 99 | 252 |
| Howell *et al*[31] | 341 | 34 | 83 | 167 |
| Varga *et al*[29] | 193 | 88 | 125 | 230 |
| Neal *et al*[25] | 221 | 56 | 144 | 264 |
| Din *et al*[24] | 500 | 54 | 149 | 263 |
| Goldschmidt *et al*[30] | 107 | NR | 341 | NR |
| Swann *et al*[28] | 202 | 24 | 53.5 | 107.5 |
| Weighted estimate | 5395 | 33.3 | 108.6 | 241.7 |
| **Time to diagnosis** | | | | |
| Howell *et al*[31] | 341 | 84 | 163 | 306 |

NR, not reported.

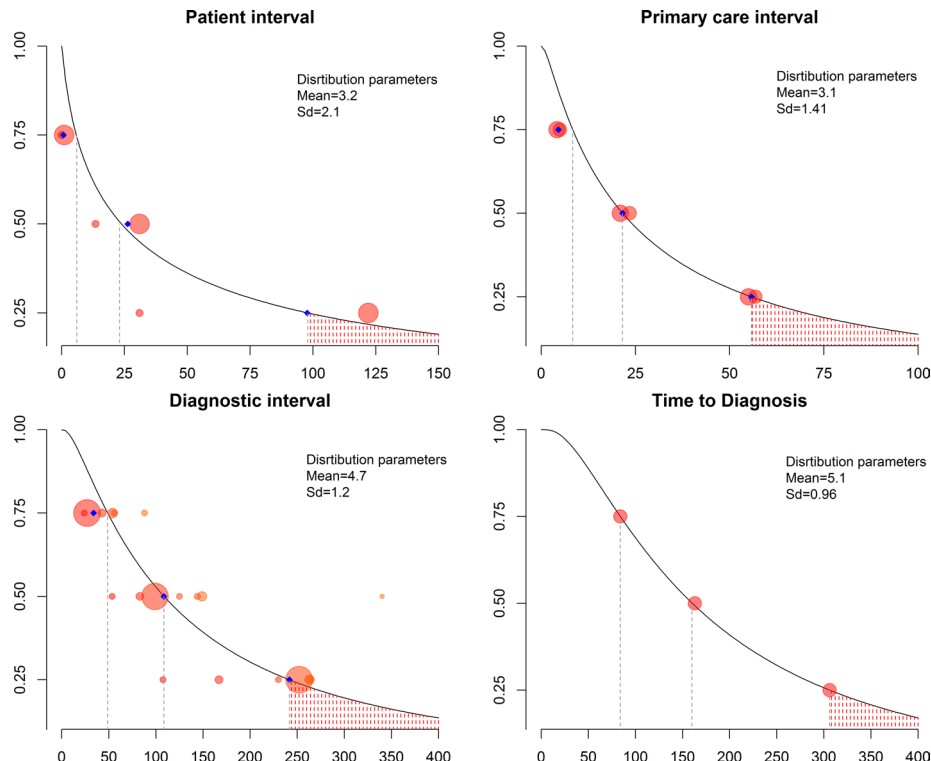

**Figure 3** Distribution of the intervals. Legend: each circle corresponds to one study, and the size is proportional to the total sample size. The blue diamond corresponds to the weighted estimate. For intervals with only one study (time to diagnosis), no weighted estimates were calculated. Y-axis corresponds to 1–probability (interval>number of days), that is 0.25 corresponds to the 75th percentile and 0.75 to the 25th percentile. X-axis corresponds to number of days.

## DISCUSSION

Our results show that patients with myeloma experience symptoms for a median of approximately 1 month before seeking help, and 25% of patients wait for more than 3 months (98 days). After attending primary care with symptoms, the median time to diagnosis is 108.6 days (IQR: 33.3–241.7) with 25% of patients waiting longer than 8 months. No studies report the referral to diagnosis interval

### Strengths and weaknesses

This is the first systematic review that quantified the patient pathway of myeloma from onset of first symptom to diagnosis. There were no restrictions in the search strategy in terms of study design or healthcare systems. We focused our search on two medical databases that are more likely to contain papers on diagnostic pathways, but we acknowledge that this might have affected the identification of all literature. To counter this, we included additional strategies like citation searching of some key references and searching the reference lists of similar reviews.

We excluded conference abstracts, although there were several that addressed the review question. The reason for this is that the length of the different intervals reported is affected by design decisions like the choice of when to start the measurement period, initial symptoms, data collection methods and so on. Conference abstracts do not report this level of detail in their methods and therefore could not be included.

In addition to measures of central tendency like the median, we included the 25th and 75th percentiles, which are particularly important since time interval data are skewed to the right. Examining all three can provide a more complete idea of the delays that the patients experience especially at the tails of the distribution. Measures of central tendency like the mean might not be the most appropriate to describe the distribution as they tend to be overestimated when used on positively skewed distributions which also makes comparison with other cancer intervals more difficult as they are usually quantified using the median and IQR.

The main limitation of the analysis is that currently there are no universally accepted formal methods to perform meta-analysis of medians and IQRs. To overcome this, we combined estimates of the percentiles after weighting them based on the sample size: a method equivalent to a fixed effects meta-analysis. Our estimates of the diagnostic interval might therefore be an underestimation due to the fact that the biggest study reported one of the lowest diagnostic intervals.[8] This however does not change the interpretation of the results as these estimates still suggest very long diagnostic intervals for patients with myeloma. We were not able to produce a CI around the median and IQR as these are not usually measures that are reported by the included studies; thus, we present only the point estimate of each percentile. In addition, there are no formal ways of

estimating statistical heterogeneity when meta-analysing median and IQR.

No studies reported the referral to diagnosis interval, and it was inferred based on the results of the other interval so our results regarding this interval should be interpreted with caution.

## Sources of heterogeneity

As mentioned in the strengths and limitations, no formal ways of estimating heterogeneity currently exist when performing meta-analysis of medians and IQR. In order to get an approximate measure of heterogeneity, we also performed a meta-analysis of the means for which we had CIs or we could approximate (online supplementary appendix A5), which resulted in an $I^2$ statistics of 98.6% (diagnostic interval). Although we expect high heterogeneity due to various design decisions that are described below, this statistic should be interpreted with caution as it might be an overestimation. We believe that to be the case because of the very small uncertainty for each within-study estimate. This results in very narrow CIs around each study that do not overlap and thus artificially inflate the $I^2$ statistic. In addition for three out of seven studies, either the means or the CIs had to be approximated, which could potentially be introducing more bias on the effect and heterogeneity estimates. Heterogeneity estimates might have been different if we were able to obtain CIs around median and IQR. We believe that clinical heterogeneity is more important in this case.

In order to compute intervals, the definition of the beginning and the end of the interval is crucial. There was variability in how studies defined starting points, especially for the first symptom and the first presentation to healthcare, using medical records or patient recall. Studies that use patient reported outcomes tend to suffer from recall bias, which might lead to overestimation or underestimation of the different intervals, while studies using medical records tend to suffer from loss to follow-up and misclassification.

For studies that used electronic health records, there was no agreement on when exactly myeloma starts to manifest. The time used to detect related symptoms prior to diagnosis spanned from 1 to 3 years. Although most studies used 1 year before diagnosis, there is some evidence to suggest that symptoms might be present for more than 1 year,[33] which may have led to an underestimation of intervals in these studies. However, the more you extend the symptom period, the more likely you are to detect symptoms that are unrelated to myeloma, which leads to the overestimation of the length of the intervals, especially with symptoms that are so non-specific such as back pain. In order to explore this, we conducted a sensitivity analysis where we estimated the length of the diagnostic interval by stratifying according to the time used to define the presenting symptoms with studies identifying the symptom at 1 year before diagnosis having a median of 105 days versus more than 1 year having a median of 142.3 day, which might explain some of the observed variability.

Even though there are various sources of heterogeneity, all the sensitivity analyses that were conducted did not change the result trend and their interpretation, as almost all studies reported diagnostic intervals longer than 3 months irrespectively of the way the study was conducted.

## Findings compared with existing research

Our estimate for the patient interval is in-line with the findings of another study,[22] which reported that 15% of patients with myeloma wait more than 3 months before they go to the doctor. This study explained this delay in terms of patients' lack of understanding of the seriousness of their symptoms because of their non-specific nature.

Patients with myeloma experience the longest primary care interval out of all cancers with a median of 21.6 days. Other cancers that have been shown to have long primary care intervals include renal and lung with a median of 14 days.[28] The long primary care interval for patients with myeloma could be explained by the fact that symptoms on their own are not enough for referral, and multiple blood tests need to be conducted like a full blood count, calcium, creatinine and inflammatory markers. Conducting multiple tests has been shown to extend the primary care interval.[34]

The diagnostic interval, which takes place in healthcare and could potentially be amenable to improvement, is longer for myeloma than for many other cancers. It has been shown that only 17.2% of patients with myeloma are referred through the suspected cancer referral pathway ('2 week' wait), which is lower than other cancers such as breast cancer for example (43%).[20 35] This could be due to the non-specific nature of the symptoms that make it hard for both the general practitioner (GP) and the patient to suspect the presence of myeloma. This might also explain the difference in the length of the diagnostic intervals between these two cancers as the median diagnostic interval for breast cancer is approximately 14 days.[36] Other cancers with a similarly long median diagnostic interval also have non-specific clinical presentations, such as lung cancer, which has a diagnostic interval of 88 days.[36]

## Implications for future research

We were not able to estimate the referral to diagnosis care interval directly, although it is reasonable to believe that it might be longer than the primary care. As in other cancers, referrals to different specialties or with an insufficient level of urgency or multiple referrals can prolong the referral to diagnosis interval.[37] The choice of referral route has also been shown to be a strong predictor of the length of the diagnostic interval, that is, patients that are diagnosed throught a referral pathway for cancer tend to have shorter intervals.[38] Future studies should estimate the duration of primary and referral to diagnosis intervals and investigate the impact of one setting on the other as in most cases the

type and severity of symptoms will determine the specialty and urgency of referral.

Also it is still not clear how long before diagnosis myeloma symptoms start to occur. In lung cancer, studies on symptom lead time (the time between symptoms attributable to cancer and diagnosis) show that symptom incidence increases considerably 6 months before diagnosis, but no such study has been conducted for myeloma.[39]

## CONCLUSION

Myeloma is a complex disease to diagnose due to a combination of different factors. First, myeloma symptoms (like back pain and fatigue) are common and mostly caused by benign conditions, resulting in patients not visiting their doctor. In addition the rarity of the disease, makes it hard for GPs to suspect this cancer. There is no effective screening programme for myeloma as this might result in people having a lot of unnecessary tests and potentially over diagnosing monoclonal gammopathy of undetermined significance (MGUS) thus any benefits from the screening programme cannot outweigh the cost. The reasons described above can explain why certain patients with myeloma tend to experience long diagnostic intervals. Our results indicate that in some cases the length of the diagnostic interval can be over 8 months. There is potential for meaningful reductions in the time to diagnosis especially for the diagnostic interval, which could improve patient outcomes, but more research is required in order to do that. Further and more in-depth exploration of the diagnostic pathway is required especially for the intervals we were not able to explore in this study like the referral to diagnosis interval and its link with the primary care interval. Development of interventions that aim to reduce the length of the diagnostic interval are now needed.

**Contributors** Six authors contributed to this study: CK, JO, LA, BDN, KR, AVdB. CK designed the study with input from AVdB and JO. All authors were involved in the conduct of the study, interpreting the results and in revising and correcting the manuscript. CK and LA reviewed and extracted the data from articles. CK and JO planned and conducted the analysis. The manuscript drafting was led by CK with the contribution of all authors. All authors read and approved the final version of the manuscript.

**Funding** This manuscript presents work carried out as part of a DPhil scholarship awarded to CK funded by the Primary Care Research Trust, The University of Oxford and NIHR Oxford CLAHRC. AVdB is supported through the National Institute for Health Research (NIHR) Diagnostic Evidence Co-operative Oxford at Oxford Health Foundation Trust (IS_DEC_0812_100).

**Disclaimer** This article presents independent research funded by the NIHR. The views expressed are those of the authors and not necessarily those of the NHS, the NIHR or the Department of Health.

**Competing interests** None declared.

**Patient consent** Not required.

**Provenance and peer review** Not commissioned; externally peer reviewed.

**Data sharing statement** The dataset is available on request from the corresponding author.

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
