## [Reviewer comments · BMJ Open]

ARTICLE DETAILS

TITLE (PROVISIONAL)	Quantifying intervals to diagnosis in myeloma: a systematic review and meta-analysis
AUTHORS	Koshiaris, Constantinos; Oke, Jason; Abel, Lucy; Nicholson, Brian; Ramasamy, Karthik; van den Bruel, Ann

VERSION 1 – REVIEW

REVIEWER	Tania Seale Bangor University , United Kingdom
REVIEW RETURNED	13-Oct-2017

GENERAL COMMENTS	General comments This is a relevant paper and provides new evidence to the field. The methods as detailed by the authors in the paper, however, do not give a detailed enough account to assess whether the review was conducted with robust methods. Comments regarding methods: • Databases searched (Page 5:23). The authors report two databases were searched and give no rationale for their choice or restriction. This could possibly be considered to have reduced the identification of ALL literature.• Searching (Page 5:20-51). The authors do not detail whether secondary search strategies were implemented to increase the level of identification of evidence. The use of such strategies as citation searching, reviewing grey literature, identifying conference abstracts and approaching authors of relevant abstracts are common secondary search activities and if not used the authors may wish to add a rationale as to why these were not implemented.• Exclusions (Page 5: 20-51). There are a number of studies not included in the synthesis reported in the review that report one or more of the intervals to diagnosis. This could potentially change the statistical evaluation of the findings. These include the 'patient' interval - Keeble et al. (2014), the 'primary care' interval - Lyratzopoulos et al. (2015), the 'time to diagnosis' (or as to referred to in this paper the 'total' interval) - Kariyawasan et al. (2007); Li et al. (2012). There may be reasons for not including these studies but possibly the authors should detail exclusions more fully in order to be transparent in their exclusion processes.• Within the strengths (Page 10: 18) there is acknowledgement that the comprehensive search strategy identifies all literature but possibly the authors may like to add either a rationale for why the use of the above observations were not considered or acknowledge these as limitations. Further comments • Title (Page 1:3): The title possibly leads to ambiguity from the use of the term 'diagnostic intervals' as this is latterly used to define the
--

	measured 'diagnostic interval'. The authors may wish to consider changing the title to 'Quantifying intervals to diagnosis' to reduce this possibility. This would also apply to the sub-title used in the results section (Page 7: 52) where perhaps greater clarification could be gained from the sub-title 'Definition of the intervals to diagnosis'.  • Intervals to diagnosis. The authors refer to the majority of studies defining the intervals to diagnosis in line with the Olesen et al. (2009) study or the Aarhus statement (Weller et al. 2012) (Page 8: 50). The authors throughout the article refer to some intervals that do not reflect the interval measurements reported in the Olesen or Weller publications i.e. the secondary care interval and the total interval are reported ending at the commencement of treatment. This specifically makes the reporting of the Howell et al. (2013) (Page 9:14) study ambiguous as the authors from this paper use the term Time to Diagnosis to measure the interval from first symptoms to date of diagnosis but the authors of this review report this as the 'Total' interval. The total interval and the time to diagnosis interval are different measurement in Olesen and Weller's articles. It is not clear in this article whether this is an intentional change in measuring the intervals, but this may be confusing to the reader when comparing intervals to diagnosis reported in other cancer sites. The authors may wish to consider defining the intervals alongside the Aarhus and Olesen studies as well and using comparable terms in the review. (Page 2: 43; Page 5:29 Page 22). Whilst this change in measurement does not affect quantification and weighting of the intervals included in the review, due to the lack of a measurement of the secondary care interval, the authors offer an interpreted median for the secondary care interval which would appear to be measured until the date of diagnosis and not the date of the commencement of treatment and therefore not comparable to other cancer types. Risk of bias (Page 8 :35) is also reported to be assessed by the ending of the intervals but it is not clear whether the ending of the secondary care and total interval reflects the measured Aarhus intervals or the authors intervals reported in the appendices. • Nafees et al., 2015 (Page 18:13; 19:26; 20:10; 27:15) is incorrectly cited in the appendices and I believe should be cited as Din et al., 2015, this however, is correctly referenced in the reference list. • Use of term 'metastatic' (page 12:32). The authors refer to longer intervals being associated with advanced disease and describe the reported incident of '22% of myeloma patients having metastatic disease at diagnosis compared to breast cancer patients with 8.7%'. This term is possibly not appropriate for describing advanced stage disease in myeloma where complications of the disease may be assessed to determine the 'burden of disease' or clinical parameters used to measure 'stage' as a prognostic tool. • Statistical evaluation using weighted and logmormal distribution (Page 6: 47; 56) appears an appropriate analysis but I am not an expert.
--	--

REVIEWER	Debra Howell University of York UK
REVIEW RETURNED	18-Oct-2017

GENERAL COMMENTS	The study examines an interesting and important subject. The systematic review methods appear clearly described and transparent; the intervals examined are defined and bias is assessed using a recognised checklist. There are several general limitations to the study: 1) a small number of studies exist for comparison, meaning the diagnostic interval is the only time-period
---

reported in more than 1 or 2 papers; 2) studies are very heterogeneous (different data sources, intervals examined and definitions of intervals). Specific comments are listed below:

P2: Abstract

Line 18: The latest search for studies was over 18 months ago. It would strengthen the paper to say a final search had been run to check for more recent publications.

Line 37: Could the interval components be defined briefly so the abstract stands alone?

Lines 45 and 54: Too much attention is given to the estimated secondary care interval. Data are not available and it may be better to limit inferences about this to the discussion, if at all. Also, I think the upper range of the inferred range may be incorrect (142 days) – should both the patients and primary care interval be removed to give the secondary care interval (115.7 days). Furthermore, is subtraction of median intervals acceptable statistically?

P3, line 3: do you mean driven by the 'referral speciality' and urgency of referral...?

P12: Discussion

Line 8: In terms of the long diagnostic interval compared to other cancers, I wondered whether it would be useful to discuss the two week wait (2WW) referral criteria and how patients may be unlikely to be referred via this pathway as their symptoms tend to be non-specific and GPs do not suspect cancer (nor do patients!). Also, more should perhaps be made of the fact that multiple individual referrals may be made by the GP (i.e. not onward consultant to consultant referrals) before the myeloma diagnosis is made.

Line 21: You may need to clarify that the category 'leukaemia' includes both acute leukaemias (e.g. acute myeloid leukaemia), which are aggressive and usually diagnosed within days due to symptom severity as well as the more chronic diseases (e.g. chronic lymphocytic leukaemia), which are more indolent, have a non-specific presentation and generally take longer to diagnosis. Otherwise the duration of time does not make sense clinically.

P13: line 7: This section could be more nuanced. Hospital referrals are likely to be made to the speciality that appears most appropriate, given the presenting symptoms; urgency of referral (and whether it should be a 2WW) will also depend on the type and severity of symptoms. It might not be the 'wrong' speciality or urgency, just the one that appears most appropriate based on the available evidence.

P18: Table 1

- Insert reference numbers for studies.
- Amend Nafees reference to Din.
- Make clear that the sample size relates to the number of patients with myeloma, not total number, if multiple cancers are included.
- Whether the study is myeloma specific is not as important as whether the results are presented for myeloma patients, rather than collectively.
- Footnote should be added with abbreviations used in the table.

P18, 19, 20: Tables 1, 2 and 3 could be ordered more appropriately.

REVIEWER	Muaamar Al-Gobari Institute of social and preventive medicine, Lausanne, Switzerland
REVIEW RETURNED	03-Jan-2018

GENERAL COMMENTS	Dear editer, dear Authors, Thanks for being given the opportunity to peer review this paper. I am entitled to review it from statistical point of view but I will try to comment wherever necessary. The paper addresses the time intervals to Myeloma diagnosis. The study is of importance but, from statistical point of view, it needs a revision. I reported here some important aspects to consider in an eventual revision of the manuscript. Since your search strategy retrieved 5 UK studies out of 7, it would be of interest to study the differences of diagnostic intervals between NICE-qualifying symptoms versus NICE-non-qualifying symptoms. Since you consider a potentially substantial heterogeneity, I do not recommend meta-analyzing the data the way it is done for the next reasons 1- The data are skewed and so the median cannot be treated as similar as the mean and because the distribution of the outcome is skewed, we cannot calculate the standard deviation from the interquartile range. However, we recommend that you check this paper by Hozo et al. https://bmcmmedresmethodol.biomedcentral.com/articles/10.1186/1471-2288-5-13 This paper was reviewed by Wan et al. https://bmcmmedresmethodol.biomedcentral.com/articles/10.1186/1471-2288-14-135 You will need to calculate the mean and the standard deviation to properly meta-analyze the data. You can use the Excel spreadsheet provided by the authors and it is easy to use. Additional minor points:  - We recommend uploading the protocol as an appendix (if the managing editor permits that) or upload it somewhere or try to send it to Prospero but it seems quite late at this stage. (This comment is left to the authors and so optional). - Provide references for studies mentioned in the study characteristics (page 7 line 13, 14, 21) where appropriate. Four studies....etc. - Please specify the number of the appendix and name of the file where appropriate (page 8 line 12.) - Check that at least once the abbreviations like CPRD and SEER (Surveillance Epidemiology and End Results –Medicare database) is mentioned in full at least onnce in the manuscript (on the
---

	first mention).  - Please verify whether you talk about a weighted mean or a weighted median in page 9 or elsewhere. As you know, median and IQR are reported when the data are not normally distributed. - Secondary care intervals are inferred from one study and so added clearly to the limitations after the abstract (to take into consideration the journal style) because the actual statement is unclear. We knew later from the review that no study reported secondary care intervals but what are secondary outcomes in that statement? - On page 11, this could be affected thought by the fact that...etc. Please review the English language. - If you are willing to calculate the treatment interval (time from diagnosis to treatment), that is welcome (optional). I think that will be shorter than diagnosis interval but still important to consider.
--	---

VERSION 1 – AUTHOR RESPONSE

11-Jan-2018

Dear Mr. Koshiaris:

Manuscript ID bmjopen-2017-019758 entitled "Quantifying diagnostic intervals in myeloma: a systematic review and meta-analysis" which you submitted to BMJ Open, has been reviewed. The comments of the reviewer(s) are included at the bottom of this letter. The Editorial Office have also checked your manuscript for any minor formatting issues and these will be listed at the end of this email.

The reviewer(s) have recommended revisions to your manuscript. Therefore, I invite you to respond to the reviewer(s)' comments and revise your manuscript. Please remember that the reviewers' comments and the previous drafts of your manuscript will be published as supplementary information alongside the final version.

To revise your manuscript, log into <https://mc.manuscriptcentral.com/bmjopen> and enter your Author Center, where you will find your manuscript title listed under "Manuscripts with Decisions." Under "Actions," click on "Create a Revision." Your manuscript number has been appended to denote a revision.

You may also click the below link to start the revision process (or continue the process if you have already started your revision) for your manuscript. If you use the below link you will not be required to login to ScholarOne Manuscripts.

*** PLEASE NOTE: This is a two-step process. After clicking on the link, you will be directed to a webpage to confirm. ***

https://mc.manuscriptcentral.com/bmjopen?URL_MASK=2f8d139924a94cb09b644edb848868ad

You will be unable to make your revisions on the originally submitted version of the manuscript. Instead, revise your manuscript using a word processing program and save it on your computer. Please also highlight the changes to your manuscript within the document by using the track changes mode in MS Word or by using bold or coloured text. Once the revised manuscript is prepared, you can upload it and submit it through your Author Center.

When submitting your revised manuscript, you will be able to respond to the comments made by the reviewer(s) in the space provided. You can use this space to document any changes you make to the original manuscript. In order to expedite the processing of the revised manuscript, please be as specific as possible in your response to the reviewer(s).

You will receive a proof if your article is accepted, but you will be unable to make substantial changes to your manuscript, please take this opportunity to check the revised submission carefully.

IMPORTANT: Your original files are available to you when you upload your revised manuscript. Please delete any redundant files before completing the submission.

Because we are trying to facilitate timely publication of manuscripts submitted to BMJ Open, your revised manuscript should be submitted within 28 days. If it is not possible for you to submit your revision by this date, we may have to consider your paper as a new submission.

Once again, thank you for submitting your manuscript to BMJ Open and I look forward to receiving your revision.

Sincerely,
Dr Anna Clark
Assistant Editor, BMJ Open
BMJ, BMA House, Tavistock Square, London, WC1H 9JR
E: aclark@bmj.com
W: bmjopen.bmj.com

Associate Editors Comments to Author:

This is an interesting clinical RQ. The search appears to have been well conducted and carefully written up. I think clinicians will be interested in these findings.

They need to discuss the limitation of only searching two databases. Their search is also old and needs updating.

We have discussed more extensively the limitations of only searching two databases in the first two paragraphs of the strengths and limitations sections. We have also updated our search from February 2016 to January 2018. We have identified two new papers that have been included in our data synthesis. The first one is a study conducted in Israel where they quantify the diagnostic interval and the second one are the results of the most recent National Audit of Cancer Diagnosis in Primary Care that was published in December 2017 and in which they quantify the primary and diagnostic interval.

Editors requests:

Please revise the 'Strengths and limitations' section of your manuscript. This section should relate specifically to the methods, and should not include a general summary of, or the results of, the study.

We have revised the Strengths and Limitations section where we discuss in more detail how our choice of different methods affects our results and their interpretation. We have removed some sentences that summarize/interpreting the results. We have structured our discussion section as follows: introductory paragraph briefly summarizing the results, strengths and limitations, sources of heterogeneity, findings compared to existing literature, implications for clinical practice, implications for future research and conclusion.

Reviewer(s)' Comments to Author:

Reviewer: 1

Reviewer Name: Tania Seale

Institution and Country: Bangor University , United Kingdom

Please state any competing interests or state 'None declared': I have completed a systematic review on the diagnostic journey in myeloma. This review, includes a synthesis of the intervals to diagnosis. This has been submitted as part of my PhD thesis and will be written up for publication.

Please leave your comments for the authors below General comments This is a relevant paper and provides new evidence to the field. The methods as detailed by the authors in the paper, however, do

not give a detailed enough account to assess whether the review was conducted with robust methods.

Comments regarding methods:

- Databases searched (Page 5:23). The authors report two databases were searched and give no rationale for their choice or restriction. This could possibly be considered to have reduced the identification of ALL literature.

We decided to focus only on the two medical databases that are more likely to contain information on diagnostic pathways compared to other more specific ones (like Cochrane for example). We do acknowledge that searching only two databases might have affected the identification of all literature so we discuss it in the strengths and limitations section where we explain our rationale and acknowledge this limitation.

- Searching (Page 5:20-51). The authors do not detail whether secondary search strategies were implemented to increase the level of identification of evidence. The use of such strategies as citation searching, reviewing grey literature, identifying conference abstracts and approaching authors of relevant abstracts are common secondary search activities and if not used the authors may wish to add a rationale as to why these were not implemented.

We did use some secondary strategies like citation searching mainly of some key references like the Aarhus statement and searching the reference list of other systematic reviews which were investigating a similar topic (Neal et al. 2014) although we did not report these methods. We added a sentence in the methods section briefly describing the secondary search strategies we used to make the methods more transparent. With respect to the grey literature we decided beforehand to include only full text articles in the review. Main reason was because the length of the different intervals is affected by design decisions like the choice of when to start the measurement period, symptoms used etc. Conference abstracts would not allow us to fully evaluate the study design decisions. We added the above rationale in our strengths and limitations discussion.

- Exclusions (Page 5: 20-51). There are a number of studies not included in the synthesis reported in the review that report one or more of the intervals to diagnosis. This could potentially change the statistical evaluation of the findings. These include the 'patient' interval - Keeble et al. (2014), the 'primary care' interval - Lyratzopoulos et al. (2015), the 'time to diagnosis' (or as to referred to in this paper the 'total' interval) - Kariyawasan et al. (2007); Li et al. (2012). There may be reasons for not including these studies but possibly the authors should detail exclusions more fully in order to be transparent in their exclusion processes.

The above papers were identified and excluded for the following reasons:

The Keeble et al. (2014), Lyratzopoulos et al. (2013) and Lyratzopoulos et al. (2015) papers were using data from the National Audit of Cancer diagnosis in Primary care 2009-2010 and because we wanted to avoid double entering the same patients we did not include all the intervals reported in all these papers. The table below shows the intervals reported by each paper and we explain which ones we included in the review.

	Patient interval	Primary care interval
Keeble et al. (2014)	n=127, p50: 14 (0-40)	NA
Lyratzopoulos et al. (2013)	NA	n=176, p50: 21 (5-56)
Lyratzopoulos et al. (2015)	n=124, p50: 14 (0-31)	n=124, p50: 21 (5-62)

We included the primary care interval from the 2013 paper as they were using a bigger sample size. The 2015 paper restricted the analysis on patients who had both the patient and primary care interval reported so we selected only the patient interval from that paper. The Keeble paper reported only the patient interval which is the same as the 2015 paper and we included the 2015 paper as it was a more recent publication. The intervals in all cases are the same and the differences observed are mainly due to fluctuations in the sample size.

The Kariyawasan et al. (2007) was in our inclusion list but the data were not in the correct format. The data were reported in terms of percentages instead of medians or means. We contacted the

corresponding author of the paper but we did not receive any replies so eventually we had to exclude it as it could not be used in the data synthesis.

The Li et al. (2012) paper again was considered but we excluded it as the analysis focuses only on patients that were referred for a specific complication in particular renal impairment and provides the waiting time only for patients who attended either a haematologist or a nephrologist but not in aggregate or for the other patients so we felt that this selective group of patients was not representative of the interval.

We made the exclusions more apparent in the text by adding the references of these papers into the first paragraph of the results section and briefly describe the exclusion reasons.

- Within the strengths (Page 10: 18) there is acknowledgement that the comprehensive search strategy identifies all literature but possibly the authors may like to add either a rationale for why the use of the above observations were not considered or acknowledge these as limitations.

We added more details in the methods section on why the above observations were not included in the data synthesis and acknowledge some of the limitations mentioned above.

Further comments

- Title (Page 1:3): The title possibly leads to ambiguity from the use of the term 'diagnostic intervals' as this is latterly used to define the measured 'diagnostic interval'. The authors may wish to consider changing the title to 'Quantifying intervals to diagnosis' to reduce this possibility. This would also apply to the sub-title used in the results section (Page 7: 52) where perhaps greater clarification could be gained from the sub-title 'Definition of the intervals to diagnosis'.

We thank the reviewer for the comment and we agree. The title was changed to "Quantifying intervals to diagnosis" and the sub-title in the results section was also changed.

- Intervals to diagnosis. The authors refer to the majority of studies defining the intervals to diagnosis in line with the Olesen et al. (2009) study or the Aarhus statement (Weller et al. 2012) (Page 8: 50). The authors throughout the article refer to some intervals that do not reflect the interval measurements reported in the Olesen or Weller publications i.e. the secondary care interval and the total interval are reported ending at the commencement of treatment. This specifically makes the reporting of the Howell et al. (2013) (Page 9:14) study ambiguous as the authors from this paper use the term Time to Diagnosis to measure the interval from first symptoms to date of diagnosis but the authors of this review report this as the 'Total' interval. The total interval and the time to diagnosis interval are different measurement in Olesen and Weller's articles. It is not clear in this article whether this is an intentional change in measuring the intervals, but this may be confusing to the reader when comparing intervals to diagnosis reported in other cancer sites. The authors may wish to consider defining the intervals alongside the Aarhus and Olesen studies as well and using comparable terms in the review. (Page 2: 43; Page 5:29 Page 22). Whilst this change in measurement does not affect quantification and weighting of the intervals included in the review, due to the lack of a measurement of the secondary care interval, the authors offer an interpreted median for the secondary care interval which would appear to be measured until the date of diagnosis and not the date of the commencement of treatment and therefore not comparable to other cancer types. Risk of bias (Page 8 :35) is also reported to be assessed by the ending of the intervals but it is not clear whether the ending of the secondary care and total interval reflects the measured Aarhus intervals or the authors intervals reported in the appendices.

We agree that some of the definitions might cause some confusion. We retain the definitions of the patient, primary and diagnostic interval as they are the ones reported in the Aarhus statement/Olesen papers. The intention of this review was to summarize the different intervals up until the diagnosis stage that is why we did not try to summarize the treatment interval. In addition out of all the papers we identified none was actually reporting the length of the treatment interval.

In order not to cause confusion with how the intervals are defined in the Aarhus statement/Olesen papers we rename the "total" interval to "time to diagnosis" which is the time from first symptom until diagnosis and the "secondary care interval" is renamed to "referral to diagnosis" interval to make it clear that the endpoint of this interval is the diagnosis. We believe that these intervals are being

reflected by the Aarhus statement as the checklist evaluates the dates of first symptom, presentation, referral and diagnosis which are the dates we use to define the above intervals.

- Nafees et al., 2015 (Page 18:13; 19:26; 20:10; 27:15) is incorrectly cited in the appendices and I believe should be cited as Din et al., 2015, this however, is correctly referenced in the reference list.

We amended the references to Din et al. 2015

- Use of term 'metastatic' (page 12:32). The authors refer to longer intervals being associated with advanced disease and describe the reported incident of '22% of myeloma patients having metastatic disease at diagnosis compared to breast cancer patients with 8.7%'. This term is possibly not appropriate for describing advanced stage disease in myeloma where complications of the disease may be assessed to determine the 'burden of disease' or clinical parameters used to measure 'stage' as a prognostic tool.

This is the term that they have used in the audit to describe an advanced disease. We agree that it might cause some confusion as this term was used for all cancers and it was not myeloma specific so we have removed that sentence from the text

- Statistical evaluation using weighted and lognormal distribution (Page 6: 47; 56) appears an appropriate analysis but I am not an expert.

We also provide more detail on why we have chosen this approach in our answers to reviewer 3.

Reviewer: 2

Reviewer Name: Debra Howell

Institution and Country: University of York, UK

Please state any competing interests or state 'None declared': None declared

Please leave your comments for the authors below The study examines an interesting and important subject. The systematic review methods appear clearly described and transparent; the intervals examined are defined and bias is assessed using a recognised checklist. There are several general limitations to the study: 1) a small number of studies exist for comparison, meaning the diagnostic interval is the only time-period reported in more than 1 or 2 papers; 2) studies are very heterogeneous (different data sources, intervals examined and definitions of intervals). Specific comments are listed below:

P2: Abstract

Line 18: The latest search for studies was over 18 months ago. It would strengthen the paper to say a final search had been run to check for more recent publications.

We conducted an updating of the review up until January 2018 in which we identified two new papers that met our inclusion criteria.

Line 37: Could the interval components be defined briefly so the abstract stands alone?

We have restructured the abstract in which we added brief definitions of the different intervals in order to be able to stand alone.

Lines 45 and 54: Too much attention is given to the estimated secondary care interval. Data are not available and it may be better to limit inferences about this to the discussion, if at all. Also, I think the upper range of the inferred range may be incorrect (142 days) – should both the patients and primary care interval be removed to give the secondary care interval (115.7 days). Furthermore, is subtraction of median intervals acceptable statistically?

We corrected our mistake regarding the estimation of the interval and we apologize for our mistake. We toned down the discussion on the secondary care interval and we concentrate more on the intervals for which we have data. Although subtraction might not be the best method to estimate the length of the interval in the absence of further studies it still gives the readers an idea of how long it might be. We retain it in the results section and we emphasize it in the limitations section as a result that should be interpreted with caution. We mention this as something that can be potentially explored in future research.

P3, line 3: do you mean driven by the 'referral speciality' and urgency of referral...?

This particular sentence was removed from the abstract.

P12: Discussion

Line 8: In terms of the long diagnostic interval compared to other cancers, I wondered whether it would be useful to discuss the two week wait (2WW) referral criteria and how patients may be unlikely to be referred via this pathway as their symptoms tend to be non-specific and GPs do not suspect cancer (nor do patients!). Also, more should perhaps be made of the fact that multiple individual referrals may be made by the GP (i.e. not onward consultant to consultant referrals) before the myeloma diagnosis is made.

We have expanded the discussion in order to include the points above with an emphasis to the two week referral criteria. Multiple referrals are more likely to affect the length of the referral to diagnosis interval if we use the first referral as the start of the measuring period so we mention it in the "Implications for future research" section.

Line 21: You may need to clarify that the category 'leukaemia' includes both acute leukaemias (e.g. acute myeloid leukaemia), which are aggressive and usually diagnosed within days due to symptom severity as well as the more chronic diseases (e.g. chronic lymphocytic leukaemia), which are more indolent, have a non-specific presentation and generally take longer to diagnosis. Otherwise the duration of time does not make sense clinically.

We specified in the text that the term leukaemia includes all type of leukaemia (chronic and acute)

P13: line 7: This section could be more nuanced. Hospital referrals are likely to be made to the speciality that appears most appropriate, given the presenting symptoms; urgency of referral (and whether it should be a 2WW) will also depend on the type and severity of symptoms. It might not be the 'wrong' speciality or urgency, just the one that appears most appropriate based on the available evidence.

We amended this sentence by avoiding using the word "wrong" and by adding the following concluding sentence to the paragraph: "Future studies should not only estimate the duration of primary and referral to diagnosis intervals, but also investigate the impact of one setting on the other as in most cases the type and severity of symptoms will determine the speciality and urgency of referral".

P18: Table 1

- Insert reference numbers for studies.
Fixed
- Amend Nafees reference to Din.
Fixed
- Make clear that the sample size relates to the number of patients with myeloma, not total number, if multiple cancers are included.
We replaced the sample size with number of myeloma patients
- Whether the study is myeloma specific is not as important as whether the results are presented for myeloma patients, rather than collectively.
We have removed this column from the table and we just kept it in the results section.
- Footnote should be added with abbreviations used in the table.

P18, 19, 20: Tables 1, 2 and 3 could be ordered more appropriately.
We have ordered the studies in the tables in chronological order

Reviewer: 3

Reviewer Name: Muaamar Al-Gobari

Institution and Country: Institute of social and preventive medicine, Lausanne, Switzerland

Please state any competing interests or state 'None declared': None declared

Please leave your comments for the authors below Thanks for being given the opportunity to peer review this paper.

I am entitled to review it from statistical point of view but I will try to comment wherever necessary.

The paper addresses the time intervals to Myeloma diagnosis. The study is of importance but, from statistical point of view, it needs a revision.

I reported here some important aspects to consider in an eventual revision of the manuscript.

Since your search strategy retrieved 5 UK studies out of 7, it would be of interest to study the differences of diagnostic intervals between NICE-qualifying symptoms versus NICE-non-qualifying symptoms.

That would have been a very useful analysis but we do not have enough data to do that. Although we have quite a lot of studies from the UK not all of them report the diagnostic interval. Among the 7 studies that quantify the diagnostic interval (after updating the review) only 4 studies are from the UK and one of them does not report the symptoms used for myeloma. This leaves us with only 3 studies and amongst these 3 studies 2 use CPRD data which use the same symptoms.

Since you consider a potentially substantial heterogeneity, I do not recommend meta-analyzing the data the way it is done for the next reasons

1- The data are skewed and so the median cannot be treated as similar as the mean and because the distribution of the outcome is skewed, we cannot calculate the standard deviation from the interquartile range.

However, we recommend that you check this paper by Hozo et al.
<https://bmcmmedresmethodol.biomedcentral.com/articles/10.1186/1471-2288-5-13>

This paper was reviewed by Wan et al.

<https://bmcmmedresmethodol.biomedcentral.com/articles/10.1186/1471-2288-14-135>

You will need to calculate the mean and the standard deviation to properly meta-analyze the data. You can use the Excel spreadsheet provided by the authors and it is easy to use.

We thank the reviewer for the comments and the references attached.

Before start working on the review we did consider using the Hozo paper and the transformations described in the paper and then proceed with a classic fixed/random effects meta-analysis depending on the heterogeneity level but after careful consideration we decided against it for the following reasons

1. By transforming our data from median and IQR to mean and SD then we essentially lose all information about the 25th and 75th percentiles. Given that our analysis is on time intervals to diagnosis of cancer which are skewed to the right we believe that it is important to concentrate not only on measures of central tendency like the mean and the median but also investigate what happens at the end of the distribution as those patients at the end will be more likely the ones experiencing the longest delays and those are the patients who will be more likely to experience the worst outcomes. This is exactly the point we were trying to make by fitting the log normal distribution through the 3 pooled estimates.

2. The mean tends to overestimate measures of central tendency when used on right skewed distributions and it would make comparison with other cancer intervals or myeloma intervals reported in future studies difficult as most of them are quantified using the median rather than the mean.
3. We had to use the Hozo transformation in 3 out of 7 studies where the mean was not reported or we were not able to obtain additional data by contacting the authors. A comparison was conducted between the transformation and the studies that were reporting the mean and sd and we noticed that it was giving us very different results under certain circumstances (i.e. Very skewed distributions). For example the Varga paper reported the mean to be 190.8 (176.8). When we used the transformation it gave us a very different result 147.7 (106.1). The results are also affected quite a lot depending on the type of transformation that we use (3 scenarios which vary according to the data reported in the paper). By using this approach we felt that we would be introducing more bias given that we had to do apply the method in almost half the studies.
4. Our analysis is closer to a fixed effects meta-analysis which means that more weight is assigned on the bigger study. Specifically in our dataset this indicates that we might actually be underestimating the length of the different intervals but this does not change the clinical message of the paper that myeloma patients experience delays of more than 3 months. We report this into the paper
5. A sensitivity analysis using the proposed methods and random effects meta-analysis was conducted and we found that the mean time to diagnosis is 156 days compared to our estimate of 109 days which could be a factor of using the means as the preferred measure of analysis which tends to overestimate the interval, studies being assigned more uniform weights due to random effects meta-analysis and potential bias introduced by using the transformations (forest plot attached). Clinically speaking the conclusion of the paper does not change and by using our method we also have estimates for the 25th and 75th percentiles which give us a more complete idea of the delays that patients experience although they might be slightly underestimating the interval which we acknowledge in the strengths and limitations section.
6. Although we expected high heterogeneity due to the various design decisions made in the different studies which are discussed in the “Sources of heterogeneity” and can be seen in table 3 we believe that the I² estimated in the meta-analysis mentioned above might be an overestimation. Reason for that is the very small statistical uncertainty for each within study mean estimate (evident from the very narrow confidence intervals) which might be artificially inflating the I-squared statistic.

Additional minor points:

- We recommend uploading the protocol as an appendix (if the managing editor permits that) or upload it somewhere or try to send it to Prospero but it seems quite late at this stage. (This comment is left to the authors and so optional).
We have attached the protocol we used when we started the review as additional supplementary material. We leave the decision for its inclusion up to the managing editor.
- Provide references for studies mentioned in the study characteristics (page 7 line 13, 14, 21) where appropriate. Four studies....etc.
References have been included
- Please specify the number of the appendix and name of the file where appropriate (page 8 line 12.)
We specified the appendix number in methods and risk of bias sections
- Check that at least once the abbreviations like CPRD and SEER (Surveillance Epidemiology and End Results –Medicare database) is mentioned in full at least onnce in the manuscript (on the first mention).
These abbreviations are mentioned in full at the first time they are mentioned in the text
- Please verify whether you talk about a weighted mean or a weighted median in page 9 or elsewhere. As you know, median and IQR are reported when the data are not normally distributed.
We have corrected it in the text
- Secondary care intervals are inferred from one study and so added clearly to the limitations after the abstract (to take into consideration the journal style) because the actual statement is unclear. We knew later from the review that no study reported secondary care intervals but what are secondary outcomes in that statement?
We made the sentence clearer
- On page 11, this could be affected thought by the fact that...etc. Please review the English language.
We changed the language
- If you are willing to calculate the treatment interval (time from diagnosis to treatment), that is welcome (optional). I think that will be shorter than diagnosis

Unfortunately we cannot estimate the treatment interval as the included studies are reporting intervals only up to the diagnosis time point. No studies report when treatment started.

VERSION 2 – REVIEW

REVIEWER	Muaamar Al-Gobari Institute of social & preventive medicine (IUMSP) Lausanne Unviversity Hospital (CHUV) Switzerland
REVIEW RETURNED	19-Feb-2018

GENERAL COMMENTS	The author has satisfactorily replied to previous comments. I recommend the publication of the paper. Some more comments to the author: Heterogeneity is a real concern in pooling the studies/the data. You already tried to explain the observed variability between and within studies. Although you might consider more analysis like comparing medical records to patient-reported outcomes (hyphenated as patient-reported data...etc.), this would not change the current result trend. Please acknowledge the I-square statistic in the “Sources of Heterogeneity” and your interpretation as written in the author’s reply
--

	to reviewers. Page 7: please provide references for “only 1 or 2 papers”. That is all. Thank you.
--	---

REVIEWER	Tania Seale North Wales Centre for Primary Care Research Bangor University
REVIEW RETURNED	22-Feb-2018

GENERAL COMMENTS	This is an important and valuable manuscript. Substantial amendments and improvements have been made. I have only one minor clarification. The authors have updated their searchers and identified the Goldschmit, et al. study and included it in this version of the manuscript. The authors clarify studies are excluded were the participants had asymptomatic forms of myeloma (MGUS or asymptomatic myeloma)but include studies where participants have a diagnosis of multiple myeloma (MM). The Goldschmit article reports 110 cases where they define 103 cases of MM and 7 cases of plasmacytoma but report outcomes for the whole 110 participants. The study does not fully report whether these 7 participants had a solitary plasmacytoma but i think possible this could be inferred. The authors performed a sensitivity analysis for the diagnostic interval and excluded the Goldschmit study as an outlier. I wonder whether the inclusion of the plasmacytoma cases is the reason for this and whether for clarity further definition of the different sample populations in this study would be beneficial?
---

REVIEWER	Debra Howell, Senior Research Fellow Epidemiology and Cancer Statistics Group Department of Health Sciences University of York York YO10 5DD
REVIEW RETURNED	12-Mar-2018

GENERAL COMMENTS	Thank you for asking me to review this paper. Comments are as follows: I'm not sure whether too much has now been made of the referral interval, particularly as it could not be examined as part of the review. Mention of this interval appears somewhat repetitive. P43 paragraph 2 (line 28). This section could be described more clearly. I think the important issue is that some studies determined the onset of myeloma from medical records based on pre-selected symptoms (e.g. Friese et al 2009). Others based the onset of myeloma on the date of the patients' first self-reported symptom, and also documented the full range of symptoms reported by patients during this time (e.g. Howell et al, 2013). P43 paragraph 3 (line 51). 'identified the first symptom...' – this is a little confusing. It may be clearer to explain that authors used various pre-diagnostic time-intervals to identify the first symptoms (1, 2 or 3 years). P48 line 42. I would remove the reference to leukaemia here. The category is not very meaningful as it includes both acute disease (e.g. AML, which is acute in onset and generally diagnosed very
--

	quickly) and indolent disease (e.g. CLL, which presents more gradually and may take a long time to diagnose). P49 line 42. The meaning of this sentence is unclear. p50 line 11. The conclusion seems a little simplistic. The complexity of myeloma diagnosis does not seem to have been acknowledged (i.e. symptoms commonly seen in benign, self-limiting illnesses which may be vague and gradual in onset; lack of effective screening without alarming people unnecessarily if they are found to have MGUS; normal blood tests until end organ damage is present. In this context, it is challenging for patients and GPs to recognise symptoms indicative of myeloma, particularly as there is little awareness of this disease among the general public and GPs see so few people with this cancer. Although the standard of written English is acceptable, there are places where it could be improved (e.g. p37 line 46 'no studies described...'; p41 line 46 'more than one study was available...'. There are also a couple of very long sentences (e.g. p41 line 25; p49 line 10).
--	--

VERSION 2 – AUTHOR RESPONSE

We thank the reviewers for their careful review and constructive suggestions. We believe that the manuscript has been improved after implementing the reviewer suggestions/edits. Please find our responses to the comments below.

Reviewer(s)' Comments to Author:

Reviewer: 3

Reviewer Name: Muaamar Al-Gobari

Institution and Country: Institute of social & preventive medicine (IUMSP), Lausanne University Hospital (CHUV), Switzerland

Please state any competing interests or state 'None declared': None declared

Please leave your comments for the authors below

The author has satisfactorily replied to previous comments.

I recommend the publication of the paper.

Some more comments to the author:

Heterogeneity is a real concern in pooling the studies/the data. You already tried to explain the observed variability between and within studies. Although you might consider more analysis like comparing medical records to patient-reported outcomes (hyphenated as patient-reported data...etc.), this would not change the current result trend. Please acknowledge the I-square statistic in the "Sources of Heterogeneity" and your interpretation as written in the author's reply to reviewers.

We have acknowledged the I-square statistic by inserting the adding the following

"Sources of heterogeneity section"

As mentioned in the strengths and limitations no formal ways of estimating heterogeneity currently exist when performing meta-analysis of medians and IQR. In order to get an approximate measure of heterogeneity we also performed a meta-analysis of the means for which we had confidence intervals or we could approximate (appendix A4) which resulted in an I-squared statistics of 98.6% (diagnostic interval). Although we expect high heterogeneity due to various design decisions which are described below, this statistic should be interpreted with a lot of caution as it might be an overestimation. We believe that to be the case because of the very small uncertainty for each within-study estimate. This results in very narrow confidence intervals around each study which do not overlap and thus artificially inflate the I-squared statistic. In addition for three out of seven studies the means or the confidence

intervals had to be approximated which could potentially be introducing more bias on the effect and heterogeneity estimates. Heterogeneity estimates might have been different if we were able to obtain confidence intervals around median and IQR. We believe that clinical heterogeneity is more important in this case.

We have also added the following concluding sentence in sources of heterogeneity section: Even though there are various sources of heterogeneity all the sensitivity analyses that were conducted were not changing the result trend and their interpretation as almost all studies were reporting diagnostic intervals longer than three months irrespectively of the way the study was conducted.

We also provided the forest plot for the pooled mean in the appendix

Page 7: please provide references for "only 1 or 2 papers".

We have referenced table 1 in that sentence with study characteristics which illustrates in detail the intervals that are reported by only one or two studies.

That is all. Thank you.

Reviewer: 1

Reviewer Name: Tania Seale

Institution and Country: North Wales Centre for Primary Care Research, Bangor University

Please state any competing interests or state 'None declared': I have an unpublished pending systematic review mapping diagnostic journey's in myeloma

Please leave your comments for the authors below

This is an important and valuable manuscript. Substantial amendments and improvements have been made. I have only one minor clarification. The authors have updated their searchers and identified the Goldschmit, et al. study and included it in this version of the manuscript. The authors clarify studies are excluded were the participants had asymptomatic forms of myeloma (MGUS or asymptomatic myeloma)but include studies where participants have a diagnosis of multiple myeloma (MM). The Goldschmit article reports 110 cases where they define 103 cases of MM and 7 cases of plasmacytoma but report outcomes for the whole 110 participants. The study does not fully report whether these 7 participants had a solitary plasmacytoma but i think possible this could be inferred. The authors performed a sensitivity analysis for the diagnostic interval and excluded the Goldschmit study as an outlier. I wonder whether the inclusion of the plasmacytoma cases is the reason for this and whether for clarity further definition of the different sample populations in this study would be beneficial?

Although it is possible those plasmacytoma patients might be affecting the estimates it still cannot explain the difference observed between this study and the rest as plasmacytoma is only present in 7 patients (around 6%) which should not be affecting the estimates to such an extent. In addition in table 1 of that paper they present the time to diagnosis based on the DSS and ISS staging (suggesting that these MM patients) which varies from 11.9 to 13.5 and is very similar with the estimate of the 107 patients for which they had sufficient data to estimate the interval (11.2 months). We did clarify the different populations in this study by adding more information in table 1 where we describe the sample size by adding a footnote with a break-down of the population.

Reviewer: 2

Reviewer Name: Debra Howell, Senior Research Fellow

Institution and Country: Epidemiology and Cancer Statistics Group, Department of Health Sciences, University of York, York, UK

Please state any competing interests or state 'None declared': None declared

Please leave your comments for the authors below Thank you for asking me to review this paper.
Comments are as follows:

I'm not sure whether too much has now been made of the referral interval, particularly as it could not be examined as part of the review. Mention of this interval appears somewhat repetitive.

We have excluded most of our inferences regarding the length of the referral to diagnosis due to lack of studies that quantify it and to avoid repetition. We have kept in our discussion the impact that different referral choices might have on the diagnostic interval due to the fact that the diagnostic interval is a function of delays that are observed in the primary and the secondary care setting and the referrals play a big part in that. We mention the length of the referral to diagnosis interval in the Implications for future research section and in our conclusion as something that needs to be explored but we avoid making any conclusions on its length given the lack of data.

P43 paragraph 2 (line 28). This section could be described more clearly. I think the important issue is that some studies determined the onset of myeloma from medical records based on pre-selected symptoms (e.g. Friese et al 2009). Others based the onset of myeloma on the date of the patients' first self-reported symptom, and also documented the full range of symptoms reported by patients during this time (e.g. Howell et al, 2013).

We think that Friese et al 2009 and the CPRD studies use a very similar way to define the onset of symptoms. For Friese it was the first claim for a sign or symptom within the year before diagnosis from a predefined list (they use ICD-9 codes) and for CPRD is the same procedure but they use more symptoms and they identify symptoms with the use of READ codes. It is indeed different compared to Howell et al. 2013 where the full range of symptoms is documented. We added the following sentence at the end of the first paragraph in this section for clarity:

"In addition studies using CPRD or SEER data were using predefined symptoms lists to identify the onset of disease while other studies like Howell et al. documented the full range of symptoms reported by the patients during this time."

P43 paragraph 3 (line 51). 'identified the first symptom...' – this is a little confusing. It may be clearer to explain that authors used various pre-diagnostic time-intervals to identify the first symptoms (1, 2 or 3 years).

We have changed the sentence

"The authors of the studies used various pre diagnostic time intervals to identify the first symptom (at one, two or three years before diagnosis). Three identified the first symptom at one year before diagnosis, three at two years and one at three years. One study used patient reported dates."

P48 line 42. I would remove the reference to leukaemia here. The category is not very meaningful as it includes both acute disease (e.g. AML, which is acute in onset and generally diagnosed very quickly) and indolent disease (e.g. CLL, which presents more gradually and may take a long time to diagnose).

We have removed the reference to leukaemia and we have kept only the lung cancer one.

P49 line 42. The meaning of this sentence is unclear.

We have clarified the sentence:

"The choice of referral route has been shown to be a strong predictor of the length of the diagnostic interval i.e. patients that are diagnosed through a referral pathway for cancer tend to have shorter intervals"

p50 line 11. The conclusion seems a little simplistic. The complexity of myeloma diagnosis does not seem to have been acknowledged (i.e. symptoms commonly seen in benign, self-limiting illnesses which may be vague and gradual in onset; lack of effective screening without alarming people unnecessarily if they are found to have MGUS; normal blood tests until end organ damage is present. In this context, it is challenging for patients and GPs to recognise symptoms indicative of myeloma, particularly as there is little awareness of this disease among the general public and GPs see so few people with this cancer.

Thank you, this is a fair point. We changed the discussion/conclusion to the following

“Myeloma is a complex disease to diagnose due to a combination of different factors. Firstly, myeloma symptoms (like back pain and fatigue) are common and mostly caused by benign conditions, resulting in patients not visiting their doctor and in combination with the rarity of the disease, making it hard for GPs to suspect this cancer. In addition there is no effective screening as this might result in people having a lot of unnecessary tests and potentially over diagnosing MGUS thus any benefits from the screening programme cannot outweigh the cost. Due to the above myeloma patients tend to experience long diagnostic intervals and our results indicate that in some cases it can be over eight months. There is potential for meaningful reductions in the time to diagnosis especially for the diagnostic interval which could improve patient outcomes but more research is required in order to do that. Further and more in-depth exploration of the diagnostic pathway is required especially for the intervals we were not able to explore in this study like the referral to diagnosis interval and its link with the primary care interval and development of interventions that aim to reduce the length of the diagnostic interval.

Although the standard of written English is acceptable, there are places where it could be improved (e.g. p37 line 46 ‘no studies described...’; p41 line 46 ‘more than one study was available...’. There are also a couple of very long sentences (e.g. p41 line 25; p49 line 10).

We have changed these sentences:

P 37 line 46 changed to:

“No studies reported data for the referral to diagnosis interval”

P41 line 46:

“For intervals reported by more than one study, the pooled estimate was calculated by taking a weighted mean for each percentile and the weight was obtained by dividing the sample size in each study with the total numbers of patients.”

P41 line 25:

We made the sentence smaller and clearer

“The Aarhus checklist is a 20 item tool to help researchers design and evaluate studies on early diagnosis of cancer. It assesses risk of bias such as time point measurement and interval definition.

P49 line 10:

This was one of the sentences removed in order to moderate our conclusions regarding the referral to diagnosis interval